# Compression and Fungal Heat Production in Maize Bulk Considering Kernel Breakage

Chaosai Liu [1], Yang Zhou [1,2,3,*], Guixiang Chen [1,2,3,*], Deqian Zheng [1,2,3] and Longfei Yue [1]

1    College of Civil Engineering, Henan University of Technology, Zhengzhou 450001, China; lcs@stu.haut.edu.cn (C.L.); deqianzheng@haut.edu.cn (D.Z.); yuelongfei1988@163.com (L.Y.)
2    Henan Key Laboratory of Grain Storage Facility and Safety, Zhengzhou 450001, China
3    Henan International Joint Laboratory of Modern Green Ecological Storage System, Zhengzhou 450001, China
*    Correspondence: robertzhouy@haut.edu.cn (Y.Z.); cgx@haut.edu.cn (G.C.)

**Abstract:** Breakage in maize kernels and vertical pressure in grains lead to the uneven distribution of grain bulk density, which easily causes undesired problems in terms of grain storage. The objective of this study was, therefore, to determine the compression and heat production of the whole kernel (WK) and half kernel (HK) under two different loadings, i.e., 50 and 150 kPa, in maize bulk. An easy-to-use element testing system was developed by modification of an oedometer, and an empirical–analytical–numerical method was established to evaluate fungal heat production, considering kernel breakage and vertical pressure. Based on the experimental results, it was found that breakage induced larger compression; the compression of HK was 62% and 58% higher than that of WK at 50 kPa and 150 kPa, respectively. The creep model of the Hooke spring–Kelvin model in series can be used to accurately describe the creep behavior of maize bulk. Fungi and aerobic plate counting (APC) were affected significantly by the breakage and vertical pressure. APC in HK was 19 and 15 times that of WK under 150 and 50 kPa, respectively. The heat released by the development of fungi was found to be directly related to the APC results. The average temperatures of WK and HK under 150 and 50 kPa were 11.1%, 9.7%, 7.9%, and 7.6% higher than the room temperature, respectively. A numerical method was established to simulate the temperature increase due to fungi development. Based on the numerical results, heat production ($Q$) by fungi was estimated, and the results showed that the $Q$ in HK was 1.29 and 1.32 times that of WK on average under 150 and 50 kPa. Additionally, the heat production results agreed very well with the APC results.

**Keywords:** maize bulk; kernel breakage; vertical pressure; deformation; heat production

## 1. Introduction

Maize is among the major cereal crops, with a wide planting range and high harvest in China. Over 40 million hectares were sown, producing 260 million tons of maize in 2020 [1]. However, despite a massive increase in maize production, postharvest losses of maize during storage remain a significant challenge [2]. A considerable amount of maize kernel breakage is easily caused by postharvest processes such as transportation and grain loading. It reduces the quality grade of the grain and also increases the risk of grain security. The density change due to the self-weight of maize bulk further induces variation in grain properties, such as non-uniform airflow [3] and heat conductivity rate, and affects mildew on grain.

Broken kernels are more susceptible to the development of fungi, and they produce a considerable amount of heat because the physical integrity of kernels is compromised [4–7]. The distribution of fungi in broken kernels was preliminarily studied via milling and dehulling [5]. Pietri et al. [8] and Burger et al. [9] found that fungi are usually concentrated in the bran, and the damage to kernel brans often increases the risk of fungi infection. Oliveira et al. [7] analyzed the relationship between the physical properties of maize

kernels and fungi infection and proposed that maize with a softer endosperm can present higher contamination by fungi. Coradi et al. [6] milled and separated breakage maize kernels into different particle sizes and concluded that aerobic plate counting (APC) was inversely proportional to the results of density and particle size.

Fungal growth can decompose organic matter in maize, thereby developing heat energy, especially in grain bulk with suitable temperature, moisture content, and oxygen concentration [10,11]. At a given height of grain bulk, the grain bulk will creep with storage time. The creep behavior will significantly influence the maize quality during and after storage [12]. However, the creep behavior of harvested maize grains during storage was rarely reported, especially the grain bulk deformation, considering broken kernels. In the food industry, creep behavior is mainly used to select high-quality raw materials and develop new products [13]. Sheng et al. [14] only established a creep model to describe the maize kernel, but there are few reports on the testing of maize bulk deformation. Thermal conductivity increased linearly with the bulk density of maize because the contact area between kernels influences heat transfer [15,16], resulting in the redistribution of moisture content and heat in the grain bulk [17,18]. The moisture–heat coupling model of grain bulk was established based on the principle of local heat and mass balance [19,20], but the effect of fungal heat production on the temperature of grain bulk was not considered. Recently, Wu et al. [21] proposed a solution for heat production in grain bulk, considering fungal activities induced rise in temperature, heat loss during conduction and convention, and water evaporation, but the effect of vertical pressure and kernel breakage on fungal heat production was not considered.

In this research, the authors developed an inexpensive and easy-to-use element testing system to study the effects of vertical pressure on compression and heat production. First, vertical pressure was applied to whole kernels (WKs) and half kernels (HKs) at 50 kPa and 150 kPa, respectively. The creep behavior was then investigated and analyzed using the model of Hooke spring–Kelvin model in series. Second, the temperature changes and APC were measured to study the fungi in the grain during compression. A newly empirical–analytical–numerical method was established to analyze the heat generated by fungi in grain bulk.

## 2. Materials and Methods

### 2.1. Test Materials

The maize used in this experiment was a normal, Zhengdan 958 hybrid maize with the conventional tillage, harvested in 2021, and stored in silos. Samples of maize kernels were collected directly from the silo. In total, 10 kg of the sample was taken at 20 random sampling points and sealed in a sterile sampling bag. The maize sample was filtered through a circular sieve to remove broken kernels and foreign impurities. Four maize samples were randomly selected from the filtered grains, and the mass of each sample was 400 g. Two samples were whole kernels, as shown in Figure 1a, and the kernels of the other two samples were cut into two parts along their middle, as shown in Figure 1b. The initial moisture content of maize was determined to be 9.8%, with samples of 30 g in three replications after drying at 103 °C for 72 h, and the moisture content after the test was also determined [22]. In this study, we focused on the numerical simulation and analytical investigation of the mildew process. Thus, we increased the moisture content to about 20% to shorten the test duration, which might not be the optimum moisture content for storage in reality. To obtain the desired target moisture content, maize was rewetted by adding distilled water mixed thoroughly and then hermetically sealed in polyethylene bags and stored at 4 °C for 48 h to allow moisture equilibrium [10]. Maize was mixed well, and about 400 g was randomly drawn for each treatment. The four actual levels of adjusted moisture content of maize were 21.02 ± 0.03%, 20.97 ± 0.29%, 21.04 ± 0.06% and 20.89 ± 0.13% wet basis.

(a)    (b)

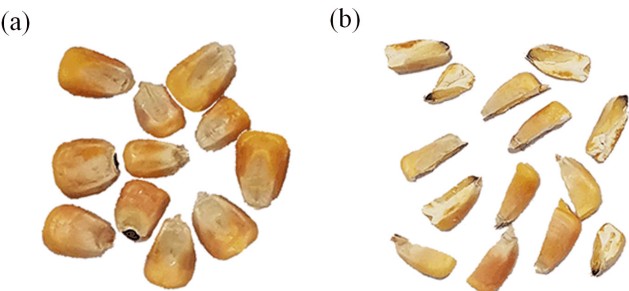

**Figure 1.** Maize kernels: (**a**) whole maize kernels and (**b**) half kernel cut along the middle.

### 2.2. Test Methods

To evaluate the effect of kernel breakage and compression on the mildew effect, a testing system was used in this study, which was modified from a conventional oedometer, normally used for soil testing, as shown in Figure 2. The testing system consists of a loading frame, a testing box, and a measuring system. The box is made of high-strength aluminum alloy and plexiglass, with a size of 120 by 120 by 55 mm. The metal loading plate is highly stiff, with a flexible rubber pad pasted below it [23]. A vertical loading screw is located in the groove so as to distribute the pressure applied to the maize bulk below. The extension length of the screw was adjusted according to the height of the maize bulk. The compression amount (*s*) of maize bulk under different vertical pressure levels and temperatures (*T*) were obtained by using a dial indicator and two T-type thermocouples (Applent Instruments Inc., Changzhou, China). The dial indicator was set on the loading screw. One thermocouple was placed in the geometric center of the maize sample, and another was attached to the external side of the box to observe the room temperature. The temperature results were recorded with an AT4508 temperature testing system (Applent Instruments Inc., China) and an automatic acquisition system.

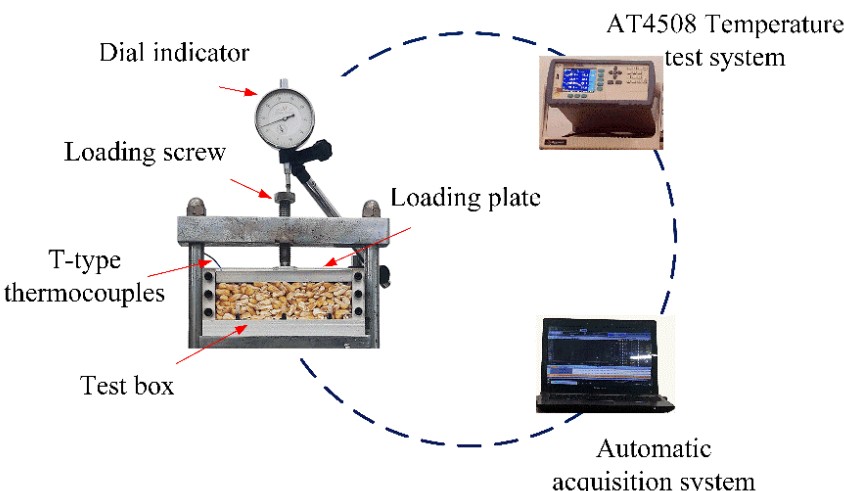

**Figure 2.** Testing system.

The walls of the testing box were coated with a thin layer of vaseline to eliminate side friction, and a 400 g sample of maize kernels was randomly taken and deposited in the testing box. As the sample area in this study was different from that of a conventional oedometer test, the vertical stress applied followed a sequence of 3, 8, 18, 28, 50, 75, 100, 125, and 150 kPa according to the calculation beforehand. In this study, two WK samples were loaded to 50 kPa and 150 kPa on multi-stages, respectively (i.e., $WK_{50}$ and $WK_{150}$), and two HK samples were loaded in the same way (i.e., $HK_{50}$ and $HK_{150}$). It is worth noting that each loading stage lasted for 20 min before reaching the final stress. Then, the four samples were continued for 7 d at constant stress. After the loading test, the APC test was performed to check mildew and grain quality.

### 2.3. Inspection of APC

Maize kernels at the center of the box, as shown in Figure 3, were selected to conduct the APC test. The sample was prepared using a rectangular plastic sampler with a size of 29 by 29 mm, which was used to take out about 30 g of maize.

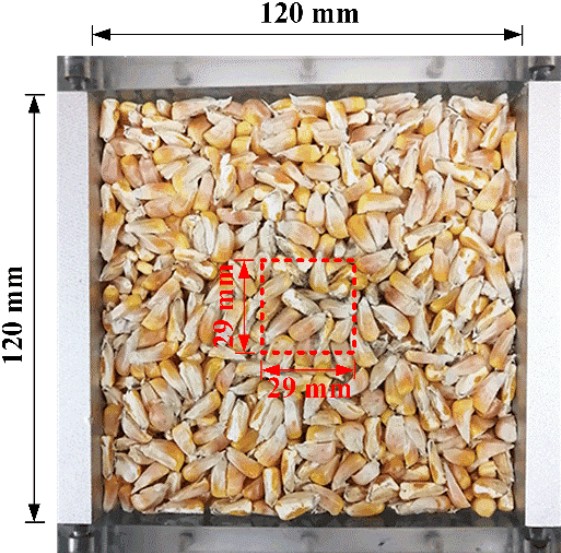

**Figure 3.** Sampling points for APC.

The specific preparation process for the APC test is as follows:

1.  Preparation of the plate counting agar (PCA): First, the tryptone, yeast extract, glucose, and agar were mixed thoroughly and dissolved fully, and the pH was controlled at $7.0 \pm 0.2$. The mixed solution was autoclaved at 121 °C for 15 min in a conical flask;
2.  Preparation of the bacterial suspension: Briefly, 25 g maize sample and 225 mL sterile water were put into a 500 mL conical flask. The conical flask was placed in an oscillator and vibrated for 30 min at a speed of 10,000 r·min$^{-1}$. The aim of this process was to fully diffuse the microorganisms on the surface of the maize kernels into the sterile water. By this process, a homogenate with a concentration of 1:10 was obtained;
3.  Preparation of homogenates with different concentrations by serial dilution: For this step, 1 mL of 1:10 sample homogenate, drawn by a 1000 µL pipette, was mixed in a test tube containing 9 mL sterile water to obtain a sample homogenate, with 1:100 concentration. The above steps were repeated several times to obtain a series of solutions with different concentrations. Normally, two homogenates with different concentrations should be prepared for later fungus culture depending on the contamination conditions. In this study, sample homogenates with concentrations of 1:10 and 1:100 were selected for maize before testing, and $10^{-5}$ and $10^{-6}$ were selected after the test;
4.  Fungus culture: Briefly, 1 mL of sample homogenate with desired concentration was taken and mixed with 15–20 mL PCA. Then, the mixture was cooled to 46 °C before injecting into 3 pre-sterilized Petri dishes. After the agar was coagulated, the dishes were turned over and cultured at $30 \pm 1$ °C for 72 h (MJPS-150, Shanghai Jing Hong Laboratory Instrument Co., Ltd., Shanghai, China);
5.  Colony counting: For each sample, one from the 3 dishes with mold colony number ranging from 100 to 150 colony-forming units (cfus) were selected, and the type and number of colonies were recorded by observing under a microscope. The number of colonies was calculated as follows:

$$N = \frac{\Sigma C}{(n_1 + 0.1 n_2)d} \tag{1}$$

where $N$ is the aerobic plate count; $\Sigma C$ is the sum of bacterial colony number; $n_1$ is the number of colonies for 1:10 homogenate; $n_2$ is the number of colonies for 1:100 homogenate; $d$ is the dilution. In this study, $d$ is equal to $10^{-1}$ for maize before the test, and $10^{-5}$ after the test [24]. The procedure for the APC test is shown in Figure 4.

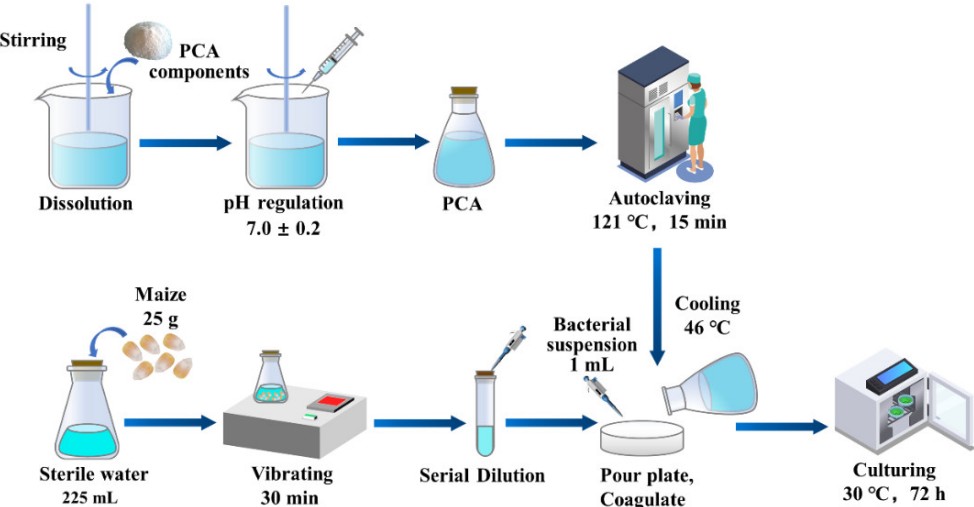

**Figure 4.** Inspection of APC.

## 3. Results and Discussion

### 3.1. Creep Behavior

Creep deformation analysis (i.e., creep) is considered a valuable method in the quality assessment of grain/food products [12]. The deformation of maize under vertical load (i.e., gravity or weight of machine) results in a reduction in void ratio and ultimately impacts the moisture content, temperature, and biochemical functions. The deformation curve of maize bulk is shown in Figure 5a. Kernel breakage and vertical pressure affected the vertical compression of maize. Compression deformations of $WK_{50}$ and $HK_{50}$ after compression were 3.73 and 6.04 cm when loading was 50 kPa, and compression deformations of $WK_{150}$ and $HK_{150}$ after compression were 6.15 and 9.72 cm when loading was 150 kPa. The vertical compression of maize bulk increased with time under constant load. The compression of WK was small under the same vertical pressure. The compression deformations of HK were 62% and 58% higher than those of WK at 50 kPa and 150 kPa, respectively. The reason is that the grain skeleton composed of WK was not easily compressed, whereas HK easily slid and filled the pores, resulting in large compression deformation.

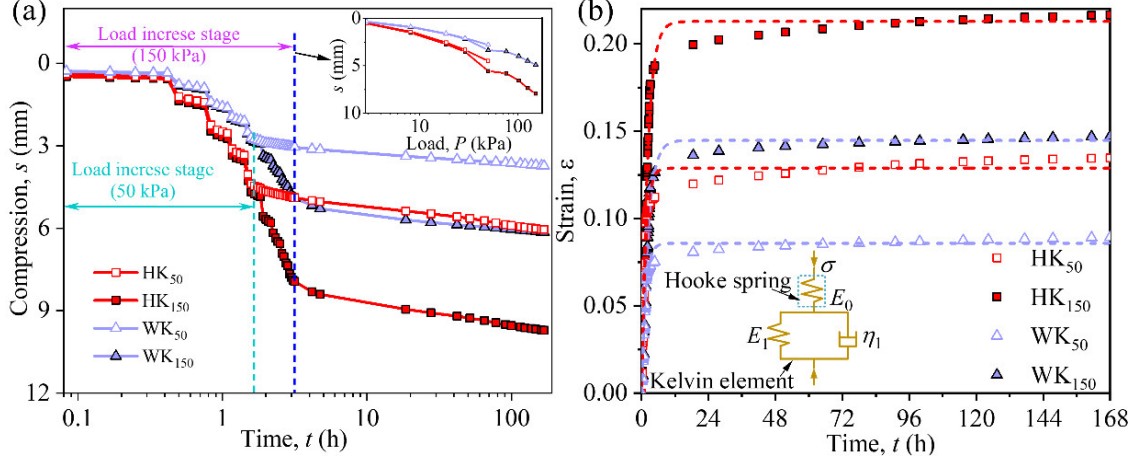

**Figure 5.** Creep behavior of maize bulk: (**a**) $s{\sim}\log(t)$ curve and (**b**) creep model.

The $s \sim \log(t)$ curve follows a reversed $s$-shape, and the deformation mainly underwent three stages: (1) Instantaneous deformation: the deformation was mainly affected by the magnitude of the load. The compression deformation occurred almost at the moment of load application, and the kernels slipped along the contact surface due to shear stress overcoming the friction at the contact surface. The kernels filled the large pores between kernels; (2) rapid deformation: the curve in this stage is mainly a section near the inflection point; the kernels were compressed, the pores between the kernels were filled, the resistance of the kernels increased, and the deformation rate decreased; (3) steady-state creep: the pores between kernels were basically filled, the resistance of kernels was greater, and the creep curve tended to be flat, roughly parallel to the abscissa axis.

Although $s \sim \log t$ can reflect the deformation trend of maize, it cannot clearly explain the internal characteristics and compression mechanism of grain, and it is not convenient to calculate the settlement of grain bulk in the process of long-term grain storage. To avoid this defect and consider the practicability of the model, a three-element model was used in this study, as seen in Figure 5b. The first region is represented by the Hooke spring unit, which shows the initiation of elastic deformation in a very short time frame, known as the instantaneous elastic deformation. The second region is represented by the Kelvin element, which indicates the retarded elastic deformation [14]. The Kelvin element is composed of Hooke spring and dashpot in parallel and can be expressed as

$$\sigma = E_1 \varepsilon + \eta_1 \dot{\varepsilon} \tag{2}$$

where $\sigma$ is the applied stress (MPa); $\varepsilon$ is the total strain (dimensionless); $E_1$ is the deformation modulus of creep stage (MPa); $\eta_1$ is the viscosity coefficient of Kelvin (MPa·h).

When the pressure is applied to the model, the spring will not be compressed immediately, due to the dashpot, and deformation gradually develops. Hooke spring is connected to the Kelvin element in series, and the creep model can be expressed as

$$\eta_1 \dot{\varepsilon} + E_1 \varepsilon = \frac{E_0 + E_1}{E_0} \sigma + \frac{\eta_1}{E_0} \dot{\sigma} \tag{3}$$

$$\varepsilon = \frac{\sigma}{E_0} + \frac{\sigma}{E_1} \left( 1 - e^{-\frac{E_1}{\eta_1} t} \right) \tag{4}$$

where $E_0$ is the instantaneous elastic modulus (MPa); $\varepsilon_0 = \sigma / E_0$; $t$ is the storage time (h).

The creep model of Hooke spring and Kelvin model in series can accurately describe the creep behavior of maize bulk under uniaxial compression. The fitting curve agreed very well with the test results, and all regression parameters had the coefficient of determination $R^2 > 0.97$, suggesting that the creep model can represent the creep behavior of maize bulk. Moreover, only three parameters—$E_0$, $E_1$, and $\eta_1$—need to be determined when the vertical stress $\sigma$, time $t$, and strain $\varepsilon$ are known; the model parameters are relatively few, which is convenient to describe the creep behavior of grain bulk.

### 3.2. Temperature and APC

Due to respiration in wet maize kernels and microorganisms, a considerable amount of heat is released, thus raising the grain temperature [11,25]; therefore, the temperature difference is a sensitive indicator of maize mold activity [11]. The temperatures of maize bulk are shown in Figure 6. By observing the room temperature, we found that heating intensity was high during daytime and low during nighttime. The temperature of maize bulk was significantly influenced by the laboratory environment and was higher than the room temperature. There were considerable differences caused by vertical pressure and kernel breakage. The temperature was the highest in $HK_{50}$ and lowest in $WK_{150}$. The average temperatures of $HK_{50}$, $HK_{150}$, $WK_{50}$, and $WK_{150}$ were 11.1%, 9.7%, 7.9%, and 7.6%, respectively, higher than the room temperature. The reason i believed to be the higher respiration process in broken kernels and sufficient oxygen between kernels under low

vertical pressure, thereby developing heat energy and leading to the rise in grain bulk temperature.

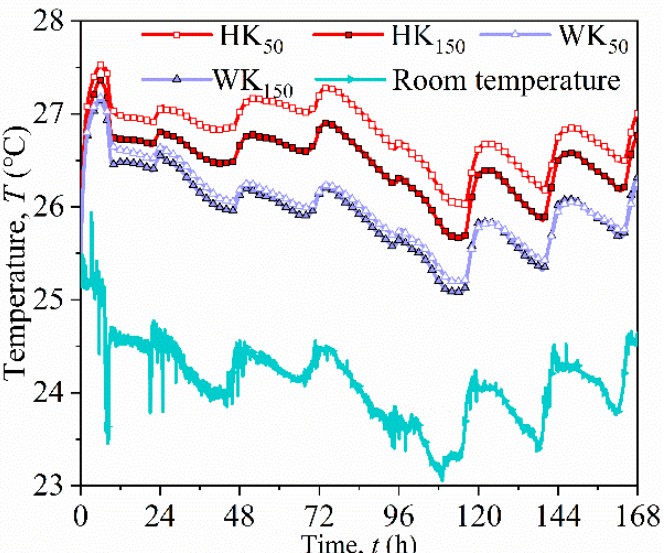

**Figure 6.** The change in maize bulk temperature.

APC of maize kernels is an important parameter to evaluate the bacterial reproductive dynamics and bacterial contamination during storage [6]. The APC values in the center of the maize sample were $3.0 \times 10^6$, $1.9 \times 10^6$, $0.2 \times 10^6$, and $0.1 \times 10^6$ cfu g$^{-1}$ for HK$_{50}$, HK$_{150}$, WK$_{50}$, and WK$_{150}$, respectively, as seen in Figure 7. The endosperm of maize had celadon hyphae and strong musty off-odor in half kernels. In addition, the color of less moldy samples was relatively bright in WK$_{150}$. These results indicated that the broken kernels were more likely to be contaminated, compared with whole kernels. The larger vertical pressure compressed and deformed the maize bulk, reducing the pores between kernels; the oxygen concentration was also low, which inhibited the development of stored grain fungi. This result corresponded very well to the temperature change shown in Figure 6. Fewer molds produced a smaller amount of respiratory heat.

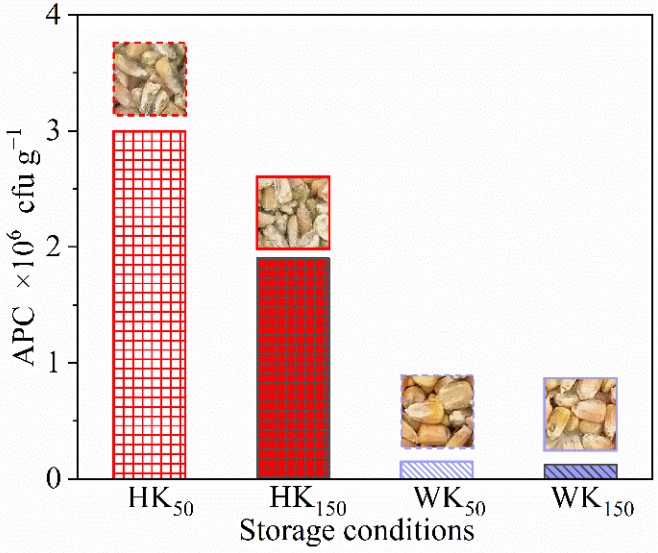

**Figure 7.** Aerobic plate count of maize stored for 7 d.

### 3.3. Physical Modeling of Heat Transfer and Verification

To investigate heat transfer due to physical factors during the testing process, finite element (FE) software COMSOL Multiphysics was used to establish numerical modeling. For simplicity, the maize kernels were assumed as porous media with isotropic properties and in local thermodynamic equilibrium with their surrounding air. Additionally, the laminar air in the void between kernels was incompressible under vertical load. However, the buoyant force and natural convection due to temperature gradient can be considered. The flow rate is assumed to be unchanged, and the 3D flow field can be expressed as

$$\frac{\partial u_j}{\partial x_j} = 0 \tag{5}$$

where $u_j$ ($j = 1,2,3$) is the air flow rate in the direction $x_j$, $u_1 = u$, $u_2 = u_3 = v$. In rectangular Cartesian coordinate system, $x_1 = x$, $x_2 = y$, $x_3 = z$. The Brinkman–Darcy formulation was incorporated with the maize domain to represent the air flow. Additionally, the momentum satisfies the Boussinesiq's approximation, then

$$\rho_a \frac{\partial u_i}{\partial t} + \frac{\rho_a u_j}{\phi} \frac{\partial u_i}{\partial x_j} = -\frac{\partial p}{\partial x_i} + \frac{\partial}{\partial x_j}\left[\mu \frac{\partial u_i}{\partial x_j}\right] + \rho_0 g \beta (T - T_0) - \frac{\phi \mu u_i}{K} \tag{6}$$

where $\rho_a$ is the density of air, $u_i$ is the speed of airflow, $t$ is the storage time, $p$ is the air pressure, $\phi$ is the porosity of maize bulk, $K$ is the permeability of maize bulk, $T$ is the temperature of the air and maize kernels, $\rho_0$ is the density of air at the reference temperature ($T_0$), $g$ is the gravity vector, $\beta$ is the coefficient of volumetric expansion of the air, and $\mu$ is the viscosity of air.

As there is a local thermodynamic equilibrium between maize kernels and the surrounding air, the governing conservation equation of thermal energy is

$$\rho_b c_b \frac{\partial T}{\partial t} + (\rho_a c_a) u_j \frac{\partial T}{\partial x_j} = \frac{\partial}{\partial x_j}\left[k_b \frac{\partial T}{\partial x_j}\right] + \rho_b h_s \frac{\partial W}{\partial t} \tag{7}$$

where $c_a$ is the specific heat of air; $\rho_b$, $c_b$, and $k_b$ are the dry density, specific heat, and the effective thermal conductivity of maize bulk, respectively; $h_s$ is the heat sorption of water on maize, and $W$ is the moisture content of maize in dry basis. Maize is a living organism with water absorption and desorption feature. Thus, the moisture balance formula is written as

$$\frac{\partial}{\partial t}\left[(\varepsilon \rho_a w) + (\rho_b W)\right] + \rho_a u_j \frac{\partial w}{\partial x_j} = \frac{\partial}{\partial x_j}\left[\frac{D_v \varepsilon}{\tau} \frac{\partial}{\partial x_j}(\rho_a w)\right] \tag{8}$$

where $\tau$ is the tortuosity factor of maize bulk, $D_v$ is a dimensionless rate coefficient for moisture exchange between air and maize kernels, and $w$ is the moisture content of water vapor in the air on a dry basis.

To verify the effectiveness of the numerical model, physical heat transfer for maize was investigated with both testing and FE methods. The test was conducted by placing the sample within the model box (including the top plate) in a freezer. Before testing, the model box, top plate, and spread-open kernels were placed in the freezer, with the door open for hours to reach a uniform initial temperature. Then, the whole maize kernels were put into the box and covered by the top plate. The tests were begun after closing the door, and the temperature of the freezer was set at 6 °C. The test lasted for 350 min only. Therefore, biochemical heat production could be ignored, and only physical heat transfer occurred during the test. Two T-type thermocouples were set at the central point of the sample and in the freezer to monitor the temperature change. The material parameters of the model box and maize used in FE analysis are listed in Table 1, and the sources of the parameters are specified as well. As seen in Figure 2, the maize bulk was basically sealed by the box

and top plate during the test. Thus, there was almost no air convection between the pores, and the maize moisture content slightly changed during the test. As a result, the flow rate of air was assumed to be zero, and water evaporation was not considered. A comparison of the tested and numerical results of $T$ at the central point is shown in Figure 8. After placing in the freezer, the temperature of the freezer $E_F$ gradually decreased from around 30 °C to 6 °C. It is worth noting that the maize tested $T$ was slightly higher than the numerical value for in the late stage, e.g., after 300 min. The reason is that the biochemical heat production is not zero even under the very low freezer temperature of 6 °C. In conclusion, the heat transfer considering various physical factors, such as air convection, water migration and evaporation, and local thermodynamic equilibrium, could be well simulated using the established FE model.

**Table 1.** The material parameters.

| Material | Property | Value |
|---|---|---|
| Aluminum alloy | Thickness ($l_A$) | 0.01 m |
| | Thermal conductivity ($k_A$) | 201 W·m$^{-1}$·°C $^{-1}$ |
| | Density ($\rho_A$) | 2720 kg·m$^{-3}$ |
| | Specific heat ($c_A$) | 90.64 J·kg$^{-1}$·°C $^{-1}$ |
| PMMA | Thickness ($l_P$) | 0.01 m |
| | Thermal conductivity ($k_P$) | 0.19 W·m$^{-1}$·°C $^{-1}$ |
| | Density ($\rho_P$) | 1180 kg·m$^{-3}$ |
| | Specific heat ($c_P$) | 1424 J·kg$^{-1}$·°C $^{-1}$ |
| Air | Air thermal conductivity ($k_a$) | 0.025 W·m$^{-1}$·°C $^{-1}$ |
| | Air density ($\rho_a$) | 1.205 kg·m$^{-3}$ |
| | Air specific heat ($c_a$) | 1006 J·kg$^{-1}$·°C $^{-1}$ |
| | Air tortuosity factor ($\tau$) | 1.2 |
| | Air viscosity ($\mu$) | $1.79 \times 10^{-5}$ Pa·s |
| Maize | Maize moisture content ($M_{maize}$) | 9.8% |
| | Maize density ($\rho_{maize}$) | 768 kg·m$^{-3}$ |
| | Maize thermal conductivity ($k_{maize}$) | 0.12 W·m$^{-1}$·°C $^{-1}$ |
| | Maize specific heat ($c_{maize}$) | 2223 J·kg$^{-1}$·°C $^{-1}$ |
| | Maize permeability ($K_{maize}$) | $1.9 \times 10^{-9}$ m$^2$ |

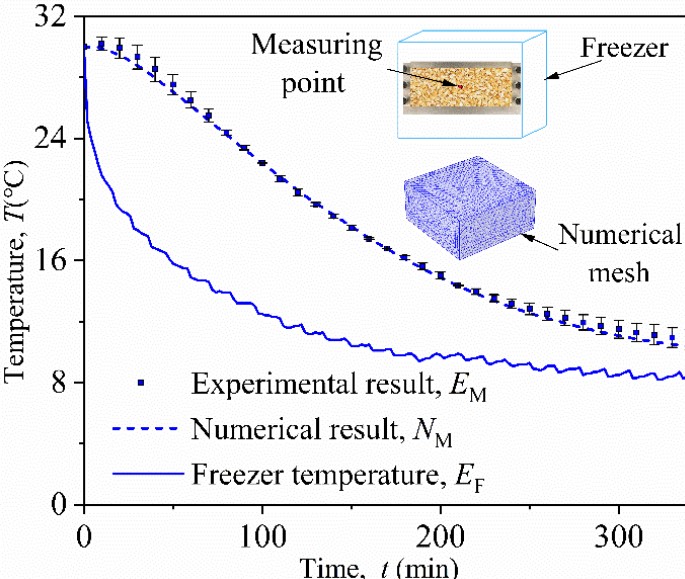

**Figure 8.** Experiment and validation of numerical model in low-temperature environment.

### 3.4. Simulation Results and Analysis

The current method normally assumed that maize is an isotropic porous medium, and the model for simulating heat and moisture transfer in grain bulk was also developed. However, the current model did not consider the effect of compression deformation on heat and moisture coupling transfer. Using the verified model mentioned in the previous section and considering vertical pressure, the variation in temperature ($T$) at the central point was simulated. The parameters involved in the FE simulation for maize are shown in Table 2. During the simulation, vertical-pressure-induced compression ($\Delta h$), considered by the input value of density ($\rho_b$), was calculated using $\Delta h$ step by step, according to the loading procedure. Similarly, the variation in porosity and thermal conductivity during the test were considered step by step as well. The experimental result and simulated result are designated as $T_E$ and $T_N$, respectively. Figure 9 presents the temperature variations in $T_E$ and $T_N$ for the four groups of maize bulk ($HK_{50}$, $HK_{150}$, $WK_{50}$, and $WK_{150}$). Clearly, it can be observed that $T_E$ is higher than $T_N$. Compared with numerical $T_N$, which only includes physical effect, the test $T_E$ resulted from physical heat transfer and biochemical effect (e.g., mildew). The temperature difference/rise actually is a consequence of heat accumulation due to a biochemical effect. Generally, half kernels under low vertical pressure, e.g., $HK_{50}$, yield larger $\Delta T$, and $WK_{150}$ yields smaller $\Delta T$. The average values of $\Delta T$ were 2.7 °C, 2.4 °C, 1.9 °C, and 1.8 °C in $HK_{50}$, $HK_{150}$, $WK_{50}$, and $WK_{150}$, respectively. This result is proportional to the APC in Figure 7, which indicates that mildew is one of the main reasons for temperature rise in grain storage. Owing to the low thermal conductivity of air between the pores of the kernels, heat conduction is slow, and the heat generated by initial fungi inside the grain bulk cannot be quickly dissipated. Increased temperature in grain bulk further provides a favorable environment for subsequent fungal growth [10,16]. This cycle of "heat conduction–temperature rise–fungal growth" accounts for the initial condition significantly. The temperature differences $\Delta T$ of HK samples were larger than those in WK, and the result is consistent with the above APC test results.

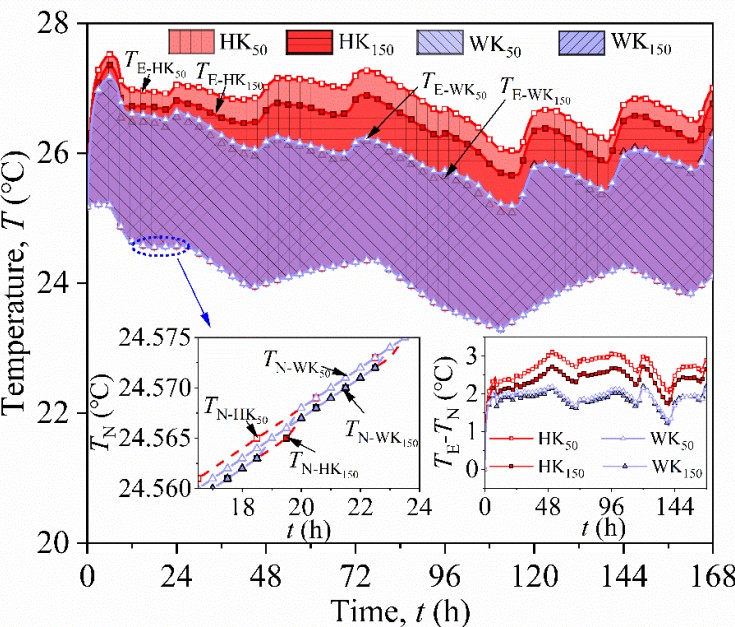

**Figure 9.** Comparison of experimental and numerical simulation results.

**Table 2.** The parameters involved in FE simulation for maize.

| Material Property | | Value |
|---|---|---|
| Moisture content ($M$) | | 21% |
| Mass ($m$) | | 0.4 kg |
| Initial height ($h_0$) | HK | 0.042 m |
| | WK | 0.045 m |
| Maize bulk density ($\rho_b$) | | $m/0.0144 \times (h_0 - s)$ |
| Particle Density ($\rho_s$) | | 1230.2 kg·m$^{-3}$ |
| Porosity ($\phi$) | | $1 - \rho_b/\rho_s$ |
| Thermal conductivity ($k_b$) | | $0.0902 + 1.165 \times 10^{-4}\,\rho_b$ [16] |

*3.5. Estimation of Fungal Heat Production*

Fungal growth decomposes organic matter in maize, thereby developing heat energy, and biochemical heat production is assumed to be heat production due to fungi development [21]. In the estimation of fungal heat production, some of the heat ($E_T$) is used to raise the temperature of the system, some heat is likely lost due to conduction ($E_C$) and convection ($E_V$), and some may be consumed during the water phase change ($E_E$), e.g., from liquid to vapor. In this study, $E_C$ was not considered, as the sample was nearly sealed. $E_E$ was also considered ignorable because the moisture content of the four samples lost only 1.1%~1.3% after being stored for 7 d. Therefore, the heat produced by fungi in the maize bulk can be determined as follows:

$$Q = f(\Delta T') = E_T + E_C \tag{9}$$

where $\Delta T'$ is the temperature difference in the maize bulk before and after fungus emergence. Heat absorbed by maize and the specific heat capacity of maize bulk were calculated using the following formula [21,26]:

$$E_T = \frac{c_b \rho_b \Delta V \Delta T'}{\Delta V} = c_b \rho_b \Delta T' \tag{10}$$

where $\Delta V$ is the spatial range affected by fungal activity (m$^3$).

The conduction heat loss ($E_C$) was determined by considering heat transfer from the geometric center of the maize sample to the wall of the testing box. The calculation space is $\Delta x = 0.12$ m, $\Delta y = 0.12$ m, and $\Delta z$ is the height of the sample at the corresponding time ($\Delta V = \Delta x \Delta y \Delta z$). Taking into account the sum of the conduction heat loss on the six sides of the testing box, according to Fourier's law, $E_C$ can be calculated as follows:

$$E_C = \frac{k_b \Delta t}{\Delta x \Delta y \Delta z} \sum_{i=1}^{6} \frac{A_i \Delta T_i}{l_i} \tag{11}$$

where $\Delta t$ is the storage time, $A_i$ is the area of one of the six faces of the testing box, and $\Delta T_i$ is the temperature difference between the center of the geometric center of the maize sample and the corresponding face $A_i$ measured at a given time $t$. $l_i$ is the distance between the center of the geometric center of the maize sample and the corresponding face $A_i$.

In this investigation, the value of temperature difference before and after fungi emergence ($\Delta T'$) can be equivalently regarded as the difference between the test and numerical results ($\Delta T$). As mentioned above, the reason for $\Delta T$ between test and simulation is whether the biochemical effect or fungi was considered. The numerical result from COMSOL Multiphysics can only include the physical factors in the simulation. Using the test and numerical temperature difference, heat production ($Q$) by fungi was evaluated, as shown in Figure 10. $Q$ was the least in WK$_{150}$, and in HK$_{50}$, HK$_{150}$, and WK$_{50}$, the values were 33.7%, 29.7%, and 1.4% higher than it on average. Additionally, it is clear that this variation in $Q$ with time agreed very well with the APC and temperature results (Figures 7 and 9).

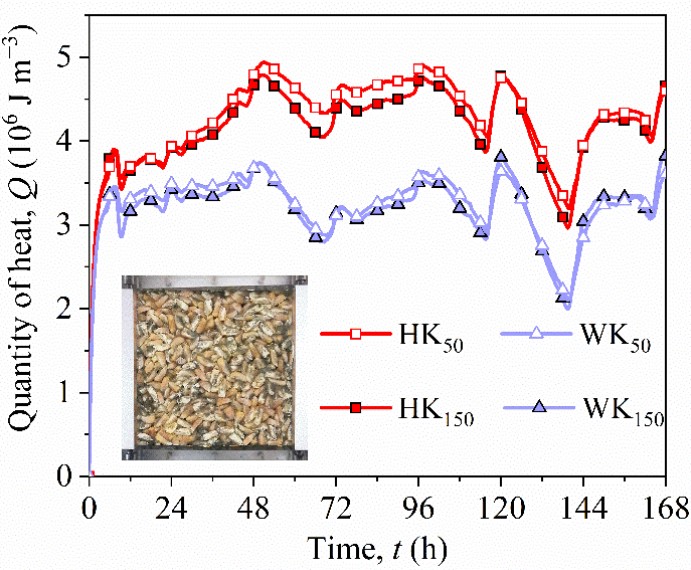

**Figure 10.** Variation in fungal heat production in maize bulk.

## 4. Conclusions

The deformation of maize under vertical load and kernel breakage of maize have an important impact on the biochemical functions of maize bulk. Maize bulk appeared to creep deformation under stable load, and the creep model of Hooke spring–Kelvin model in series can accurately describe the creep behavior of maize bulk. There was a significant difference between the compression of the whole kernel and half kernel—namely, the compression of the half kernel was 62% and 58% higher than that of the whole kernel at 50 kPa and 150 kPa, respectively. Broken kernels increase the risk of fungi infection, and deformation under vertical load results in a reduction in void ratio, thus affecting fungi development. The APC values in HK were 19 and 15 times those of WK under 150 and 50 kPa, respectively. The temperature and heat rates agreed very well with the APC results. The average temperatures of $HK_{50}$, $HK_{150}$, $WK_{50}$, and $WK_{150}$ were 11.1%, 9.7%, 7.9%, and 7.6% higher than the room temperature, respectively. The heat production rate by fungi in HK was 1.29 and 1.32 times that of WK on average under 150 and 50 kPa.

**Author Contributions:** Conceptualization, Y.Z., G.C. and L.Y.; methodology, Y.Z. and D.Z.; software, C.L.; validation, C.L.; formal analysis, C.L. and Y.Z.; investigation, C.L. and L.Y.; resources, G.C. and D.Z.; data curation, C.L. and Y.Z.; writing—original draft preparation, C.L.; writing—review and editing, Y.Z., G.C. and D.Z.; supervision, G.C. and D.Z.; project administration, D.Z.; funding acquisition, G.C. All authors have read and agreed to the published version of the manuscript.

**Funding:** This research was funded by the National Grain Public Welfare Research Project of China, grant number 201513001-01; the Innovative Funds Plan of the Henan University of Technology, grant number 2020ZKCJ05; and the Open Project of Henan Key Laboratory of Grain and oil storage facility & safety, grant number 2020KF-A02.

**Institutional Review Board Statement:** Not applicable.

**Informed Consent Statement:** Not applicable.

**Data Availability Statement:** The data presented in this study are available in the article.

**Conflicts of Interest:** The authors declare no conflict of interest.

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
