# Peer review of "Compression and Fungal Heat Production in Maize Bulk Considering Kernel Breakage"

_applsci, doi:10.3390/app12104870_

Round 1

Reviewer 1 Report

Breakage of grains during storage is one of the major post-harvest issues in maize that causes significant economic loss. The manuscript deals with a very important area of addressing the issue. The experiment was well planned and the manuscript is well presented except for few minor revisions given below:

  • What is the genetic nature of the maize hybrid ‘Zhengdan hybrid maize 958’ being used in the study? Like is it a normal or QPM maize, flint or dent maize etc., need to be clearly mentioned. As these genetic parameters also inherently influence differentially the breakage of grains during storage.
  • Was there any loss in weight of the grains observed as the applied het would have also caused reduction in moisture and eventually loss of weight.
  • Keywords: Line No. 25: Delete ‘the’
  • Introduction: Line No. 47: Expand APC as it is given for first time in the text
  • Materials and Methods: Line 84: Figure 1b is mentioned but could not locate Figure 1a.
  • No figures are available to see as a part of the manuscript
  • Line 361: Author contributions need to be explicitly presented as per the standard format of the journal.
  • Line 369-372 & 380-383: Contents presented at Line No. 380-383 need to be shifted to Line No. 369-372.

Author Response

We are grateful to reviewer#1 for his/her effort in reviewing our manuscript and his/her positive feedback. This reviewer appreciates the efforts of the authors very much. And he/she believes that the manuscript deals with a very important area of addressing the maize kernel breakage issues during storage that causes significant economic loss. The experiment was well planned and the manuscript is well presented. However, there are still some issues that can be addressed, before the accept recommendation is given.

Comment 1: What is the genetic nature of the maize hybrid ‘Zhengdan hybrid maize 958’ being used in the study? Like is it a normal or QPM maize, flint or dent maize etc., need to be clearly mentioned. As these genetic parameters also inherently influence differentially the breakage of grains during storage.

Response: Thank you very much for your careful reminder and valuable advice. Zhengdan 958 is a maize single cross bred by crossing Zheng 58 as female parent and Chang 7-2 as male parent. It has the outstanding advantages of high and stable yield, multi resistance, density tolerance and strong heterosis. It is the maize variety with the largest popularization area in China. The maize used in this experiment was a normal Zhengdan 958 hybrid maize with the conventional tillage, harvested in 2021 and stored in silos. We added:

Page 2, Lines 7778:

The maize used in this experiment was a normal Zhengdan 958 hybrid maize with the conventional tillage, harvested in 2021 and stored in silos.

Comment 2: Was there any loss in weight of the grains observed as the applied heat would have also caused reduction in moisture and eventually loss of weight.

Response: Thank you very much for your careful reminder and valuable advice. As the reviewer mentioned, because the test device cannot be completely sealed, a small amount of the moisture evaporated by the heat generated by fungi in the maize bulk. The moisture content of the sample was tested and found to be reduced by 1.1% ~ 1.3% stored for 400 h, resulting in the weight loss of the sample. The changed of water content of the four samples was very small, and the loss was basically the same, so we ignored the influence of the water phase change in the calculation process. We added statement about the weight loss to specify this point in the revised paper

Page 1011, Lines 337339:

EE was also considered ignorable because the moisture content of the four samples lost only 1.1% ~1.3% after stored for 7 d.

Comment 3: Keywords: Line No. 25: Delete ‘the’

Response: Thank you very much for your careful reminder and valuable advice. And we have deleted ‘the’ in the first keyword according to your suggestion.

Page 1, Lines 28:

Keywords: maize bulk; kernel breakage; vertical pressure; deformation; heat production

Comment 4: Introduction: Line No. 47: Expand APC as it is given for first time in the text.

Response: Thank you very much for your careful reminder and valuable advice. We have expanded APC as it is given for first time in the text according to your suggestion. And similar typos were checked by the authors.

Page 2, Lines 4749:

Coradi et al. [6] milled and separated breakage maize kernels into different particle sizes, and concluded that aerobic plate counting (APC) was inversely proportional to the results of density and particle size.

Comment 5: Materials and Methods: Line 84: Figure 1b is mentioned but could not locate Figure 1a.

Response: Thank you very much for your careful reminder and valuable advice. And the description of Figure 1a has been added in the context.

Page 2, Lines 8384:

Two samples were whole kernels, as shown in Figure 1a, and the kernels of the other two samples were cut into two parts along their middle in Figure 1b.

Comment 6: No figures are available to see as a part of the manuscript.

Response: Thank you very much for your careful reminder and valuable advice. We have submitted the figures as an attachment at the time of submission. And we rearranged the contents and have added each figure to the corresponding position in the main text according to your suggestion.

Comment 7: Line 361: Author contributions need to be explicitly presented as per the standard format of the journal.

Response: Thank you very much for your careful reminder and valuable advice. We prepared the Author's contribution section in accordance with the standard format of the journal.

Page 12, Lines 383388:

Author Contributions: Conceptualization, Y.Z., C.G., and Y.L.; methodology, Y.Z. and Z.D.; software, L.C.; validation, L.C.; formal analysis, L.C. and Y.Z.; investigation, L.C. and Y.L.; resources, C.G. and Z.D.; data curation, L.C. and Y.Z.; writing—original draft preparation, L.C.; writing—review and editing, Y.Z., C.G., and Z.D.; supervision, C.G., and Z.D.; project administration, Z.D.; funding acquisition, C.G. All authors have read and agreed to the published version of the manuscript.

Comment 8: Line 369-372 & 380-383: Contents presented at Line No. 380-383 need to be shifted to Line No. 369-372.

Response: Thank you very much for your careful reminder and valuable advice. We have revised it according to your suggestion.

Page 12, Lines 390393:

Funding: This research was funded by the National Grain Public Welfare Research Project of China, grant number 201513001-01; the Innovative Funds Plan of Henan University of Technology, grant number 2020ZKCJ05; and the Open Project of Henan Key Laboratory of Grain and oil storage facility & safety, grant number 2020KF-A02.

Reviewer 2 Report

April 14, 2022

Letter to Authors

I have read and reviewed your manuscript titled “Compression and heat production of fungi in maize bulk considering kernel breakage” (applsci-167247) submitted to APPLIED SCIENCES. Maize is one of the most important cereal around the World, undoubtedly the safety during the storage of kernels is an important issue. Manuscript mainly explores the relation among the pressure and temperature with the fungal contamination. After a deep analysis of the manuscript I think it could be published after attending the next comments.

Abstract. Should be restructured. Reduce the proportion of words utilized to describe the experiment and increase the description of the results.

Introduction. L35-38. Is it necessary to include that information? If yes, please include more information about the potential risk of COVID-19 during cereals storage? L67-76. Please rewrite the Objective to make it shorter and clearer.

Figures. Include the figures in the main text after citing each of one, it is hard to understand the manuscript without the figures.

Material and Methods. It should be included next information in the section: 1) the crop management, 2) the sampling method, 3) the sampling times; 4) the method of selection of the sample size, etc.

Results and Discussion. Line 166. There is a Chinese character. There is a wide hypothetical discussion about the results presented in Figures and the derived Models, however, there is a lack of the discussion of the results with previous published information. Some explanations about the models are too redundant and could be synthetized to make them clearer to potential readers.

Conclusions. They should not repeat all the information explained in the Results and Discussion section. Should be rewritten in just one paragraph, emphasizing the most important conclusions and recommendations DERIVED from Results. Avoid discussing the main Conclusions in this section (the discussion of conclusions can be placed in Results and Discussion section or even could be under a subtitle “Perspectives”).

Overall recommendations. There are some punctuation and grammar errors along the manuscript. Uniform the way to cite the references in the main text.

I sincerely recognize the quality of articles published in APPLIED SCIENCES, then I sincerely hope that previous comments could be helpful to improve your manuscript.

Best wishes!

Author Response

We are grateful to reviewer#2 for his/her effort in reviewing our manuscript and his/her positive feedback. The summary of our work as written by this reviewer is precise. Here below we address the questions and suggestions raised by reviewer#2.

Comment 1: Abstract. Should be restructured. Reduce the proportion of words utilized to describe the experiment and increase the description of the results.

Response: Thank you very much for your careful reminder and valuable advice. And we have rewritten the Abstract section to be more meaningful.

Page 1, Lines 1027:

Abstract: Maize kernels breakage and grain vertical pressure lead to the uneven distribution of grain bulk density, which is easy to cause undesired problems of grain storage. This objective of this study was to determine the compression and heat production of whole kernel (WK) and half kernel (HK) under two different loading, i.e., 50 and 150 kPa in maize bulk. An easy-to-use element test system was developed by modification of oedometer, and an empirical analytical-numerical method was established to evaluate heat production by fungi, considering kernel breakage and vertical pressure. Based on the experimental results, it was found breakage will induce larger compression and the compression of HK is 62% and 58% higher than that of WK at 50 kPa and 150 kPa, respectively. The creep model of Hooke spring and Kelvin model in series can be used to accurately describe the creep behavior of maize bulk. Fungi and aerobic plate counting (APC) was affected significantly by the breakage and vertical pressure. The APC in HK is 19 and 15 times of WK under 150 and 50 kPa, respectively. The heat released by the development of fungi was found to be directly relate to the APC results. The average temperatures of WK and HK under 150 and 50 kPa are 11.1%, 9.7%, 7.9% and 7.6% higher than the room temperature respectively. A numerical method was established to simulate the temperature increase due to fungi development. Based on the numerical results, the heat production (Q) by fungi was estimated and the results shown that the Q in HK is 1.29 and 1.32 times of WK on average under 150 and 50 kPa. And the heat production was agreed very well with the APC results.

Comment 2: Introduction. L35-38. Is it necessary to include that information? If yes, please include more information about the potential risk of COVID-19 during cereals storage? L67-76. Please rewrite the Objective to make it shorter and clearer.

Response: Thank you very much for your careful reminder and valuable advice. We have deleted unnecessary descriptions such as COVID-19 to accurately describe the objective of this study. And we have rewritten the objective according to your suggestion.

Page 2, Lines 6874:

In this research, the authors developed an inexpensive and easy-to-use element test system to study the effects of vertical pressure on compression and heat production. First, the whole kernel (WK) and half kernel (HK) are applied 50 kPa and 150 kPa vertical pressure, respectively. The creep behavior was investigated and analyzed using the model of Hooke spring - Kelvin model in series. Second, the temperature changes and APC were measured to study the fungi in the grain during compression. A newly empirical analytical-numerical method was established to analyze the heat generated by fungi in grain bulk.

Comment 3: Figures. Include the figures in the main text after citing each of one, it is hard to understand the manuscript without the figures.

Response: Thank you very much for your careful reminder and valuable advice. We have submitted the figures as an attachment at the time of submission. And we have added each figure to the corresponding position in the main text according to your suggestion.

Comment 4: Material and Methods. It should be included next information in the section: 1) the crop management, 2) the sampling method, 3) the sampling times; 4) the method of selection of the sample size, etc.

Response: Thank you very much for your careful reminder and valuable advice. We have supplemented the details of the experimental materials according to your suggestions.

Page 2, Lines 7783:

The maize used in this experiment was a normal Zhengdan 958 hybrid maize with the conventional tillage, harvested in 2021 and stored in silos. Samples of maize kernels were collected directly from the silo. In total, 10 kg of samples was taken at 20 random sampling point and sealed in a sterile sampling bag. The maize sample was filtered through a circular sieve to remove broken kernels and foreign impurities. Four maize samples were randomly selected from the filtered grains, and the mass of each sample was 400 g.

Comment 5: Results and Discussion. Line 166. There is a Chinese character. There is a wide hypothetical discussion about the results presented in Figures and the derived Models, however, there is a lack of the discussion of the results with previous published information. Some explanations about the models are too redundant and could be synthetized to make them clearer to potential readers.

Response: Thank you very much for your careful reminder.

  • We are very sorry for such a typo, and the manuscript has been carefully examined and revised.
  • Current method normally assumed that maize is an isotropic porous medium, and the model for simulating heat and moisture transfer in grain bulk has also been developed. But current model did not consider the effect of compression deformation on heat and moisture coupling transfer. In the simulation of this study, vertical pressure-induced compression (Δh), considered by the input value of density (ρb), was calculated using Δh step-by-step, according to the loading procedure. Similarly, the variation in porosity and thermal conductivity during test were considered step-by-step as well. We added statement about the discussion of the results with previous published information.

Page 9, Lines 302311:

Current method normally assumed that maize is an isotropic porous medium, and the model for simulating heat and moisture transfer in grain bulk has also been developed. But current model did not consider the effect of compression deformation on heat and moisture coupling transfer. Using the verified model mentioned in the previous section and considering vertical pressure, the variation of temperature (T) at central point was simulated. The parameters involved in FE simulation for maize are shown in Table 2. During the simulation, vertical pressure-induced compression (Δh), considered by the input value of density (ρb), was calculated using Δh step-by-step, according to the loading procedure. Similarly, the variation in porosity and thermal conductivity during test were considered step-by-step as well.

  • In addition, we have also condensed some explanations about the model according to your suggestions to make them easier understand to potential readers.

Page 8, Lines 255257:

The Brinkman-Darcy formulation has been incorporated with the maize domain to represent the air flow. And the momentum satisfies the Boussinesiq’s approximation, then:

Page 8, Lines 269271:

Maize is a living organism with water absorption and desorption feature. Thus, the moisture balance formula is written as:

Comment 6: Conclusions. They should not repeat all the information explained in the Results and Discussion section. Should be rewritten in just one paragraph, emphasizing the most important conclusions and recommendations DERIVED from Results. Avoid discussing the main Conclusions in this section (the discussion of conclusions can be placed in Results and Discussion section or even could be under a subtitle “Perspectives”).

Response: Thank you very much for your careful reminder and valuable advice. Your suggestion is very helpful for us to improve the quality of the paper. And we have rewritten the conclusion.

Page 1112, Lines 369381:

  1. Conclusion

The deformation of maize under vertical load and kernel breakage of maize have an important impact on the biochemical functions of maize bulk. Maize bulk appeared creep deformation under stable load, and the creep model of Hooke spring - Kelvin model in series can accurately describe the creep behavior of maize bulk. There was a significant difference between the compression of whole kernel and half kernel, where half kernel is 62% and 58% higher than that of whole kernel at 50 kPa and 150 kPa respectively. Broken kernels increase the risk of fungi infection, and the deformation under vertical load results in the reduction of void ratio, and finally impacts the fungi development. The APC in HK is 19 and 15 times of WK under 150 and 50 kPa, respectively. The temperature and heat were agreed very well with the APC. The average temperatures of HK50, HK150, WK50 and WK150 are 11.1%, 9.7%, 7.9% and 7.6% higher than the room temperature respectively. The heat production by fungi in HK is 1.29 and 1.32 times of WK on average under 150 and 50 kPa.

Comment 7: Overall recommendations. There are some punctuation and grammar errors along the manuscript. Uniform the way to cite the references in the main text.

Response: Thanks for your valuable suggestion. We apologize for the punctuation and grammar errors. We have carefully checked and revised the existing problems in the manuscript. We commissioned professional teachers from University of Manitoba at Canada to revise the manuscript to meet the requirements of this journal. At the same time, considering the editor′s suggestion, we also revised the citation way of references in the main text. We have done our best to revise the manuscript, and I hope it can meet with requirement.
